# Reflex: Real-Time Vision-Language-Action Control through Streaming Inference

**Yuanchun Guo** [1]  **Bingyan Liu** [1]

## Abstract

Flow matching Vision-Language-Action (VLA) models promise precise continuous control, but their iterative denoising nature introduces fundamental incompatibilities with real-time robotics: global timestep injection invalidates KV-caching, forcing a choice between slow $O(N^2)$ recomputation or mathematically incorrect cache reuse. We present **Reflex**, a framework that enables *real-time streaming inference* for flow matching policies by exploiting the *Timestep-Invariance Property*—that perception encoders are functionally independent of the denoising loop. Reflex partitions the attention context into static, sliding, and dynamic regions, enabling $O(1)$ incremental cache updates while preserving full-batch-equivalent attention outputs for fixed inputs. To ensure stability under continuous high-frequency inference, we introduce *AdaRMSNorm*, an adaptive normalization layer that prevents BFloat16 numerical collapse by gating on flow phase. We further maximize throughput through an *async pipeline* that decouples visual encoding from action generation, combined with *operator fusion* that reduces kernel overhead. On LIBERO and Kinetix benchmarks, Reflex achieves a 2.58× inference speedup and 50Hz stable streaming, reducing reaction latency by up to 54% and enabling efficient deployment without performance degradation.

## 1. Introduction

Vision-Language-Action (VLA) models (Physical Intelligence et al., 2025; Gemini Robotics Team et al., 2025b;a; Bjorck et al., 2025) represent the frontier of embodied intel-

[1]School of Computer Science, Beijing University of Posts and Telecommunications, Beijing, China. Correspondence to: Bingyan Liu <bingyanliu@bupt.edu.cn>.

*Proceedings of the $43^{rd}$ International Conference on Machine Learning*, Seoul, South Korea. PMLR 306, 2026. Copyright 2026 by the author(s).

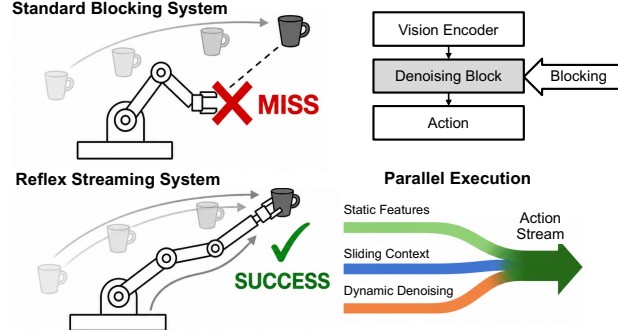

*Figure 1.* **Standard Blocking vs. Reflex Streaming.** Top: Standard flow matching inference blocks execution, leading to missed grasps on moving targets due to reaction latency. Bottom: Reflex partitions the VLA backbone into independent streams, enabling overlapping execution.

ligence, enabling robots to translate natural language instructions into continuous manipulation. Recent *Flow Matching* architectures (Lipman et al., 2023; Liu et al., 2022) have further advanced this capability by modeling high-dimensional action manifolds with greater precision than discrete tokenization (Black et al., 2025a). However, this expressiveness comes at a cost, as flow matching generates actions via an iterative denoising process that necessitates blocking execution. The robot must observe the scene, infer through the full denoising chain, and then act. In dynamic scenarios such as reaching for a moving cup, this inference latency forces the robot to react to outdated information: by the time the robot decides how to act, the target has already moved.

A common assumption is that accelerating model inference will directly yield faster robots. However, this view overlooks a critical system-level insight: the primary bottleneck is not the duration of computation, but the synchronous waiting that halts execution. As illustrated in Figure 1, standard blocking inference forces the robot to freeze while planning, causing reaction delays that lead to failures in dynamic environments. To fundamentally resolve this, the solution should lie not in faster computation, but in an asynchronous architecture that enables the robot to act while inference proceeds, which overlaps separate system stages into a continuous stream.

Asynchronous inference appears to be a promising way to reduce reaction delays (Black et al., 2025b; Ma et al., 2025; Shukor et al., 2025; Sendai et al., 2025). The main idea is to pipeline computation with execution: predicting the robot's future state to offset inference latency, and generating actions based on this forecast concurrently with the current control loop (Tang et al., 2025). This aims to make actions align with the state at execution time rather than the state observed before inference. However, while asynchronous inference removes the blocking gap between computation and execution, it does not address the computational expense of inference itself. For Flow Matching VLAs, each denoising step still needs full forward passes through the backbone, regardless of how execution is scheduled.

To make such high-frequency execution computationally feasible, standard Transformers (Vaswani et al., 2017) rely on Key-Value (KV) caching to reuse prior computations. However, this method fails for the flow matching model architectures. Because these models inject a time-dependent signal into every layer, their internal state shifts continuously during generation, which is significantly different from language models where the past context remains static. This makes cached features immediately obsolete, forcing a trade-off between the prohibitively slow re-computation and the mathematically incorrect approximation.

To address these challenges, we propose **Reflex**, a system designed to enable smooth continuous control for *Flow Matching VLAs*. Unlike prior approaches that focus on accelerating individual inference steps (Black et al., 2025b; Ma et al., 2025; Shukor et al., 2025; Sendai et al., 2025), Reflex re-architects the entire serving loop with a key insight: *the computational structure of VLA models closely aligns with the temporal structure of the physical world.*

Reflex exploits this by partitioning the model's context into static and dynamic regions. We observe that computationally intensive perception encoders process visual features that remain effectively static across consecutive control steps, while the high-speed denoising loop requires rapid updates. Based on this, Reflex introduces a specialized caching mechanism that preserves the invariant perception features while efficiently recomputing only the dynamic flow state. For a fixed observation window and fixed inputs, this reduces the complexity of each step from quadratic to constant time while preserving outputs identical to full-batch attention.

Besides latency, Reflex also tackles the distribution shift challenge inherent in long-horizon streaming. We find that continuous exposure to high-variance initialization noise, which is rarely encountered during the offline training, can destabilize standard mixed-precision inference. To prevent this, we propose AdaRMSNorm, a precision-aware normalization operator that dynamically adjusts to the variance

shifts in the streaming flow, ensuring the robot can operate indefinitely without crashing.

Finally, to bridge the remaining temporal gap between observation and actuation, Reflex incorporates a lightweight future prediction module. By forecasting the robot's state forward by the expected inference duration, it helps the policy condition on a state closer to the execution time. Experiments show that Reflex accelerates inference by $2.58\times$ and maintains performance parity on LIBERO while reducing system reaction latency by up to 54%, enabling stable, high-frequency control at 50Hz. This confirms that for embodied AI, architectural pipelining yields far greater gains than isolated component optimization. The implementation is available at `https://github.com/9yc/Reflex`.

Our contributions are:

- We introduce a fundamental shift from accelerating individual inference steps to streaming-based asynchronous execution, in order to achieve real-time VLA control.

- We present Reflex, an asynchronous serving architecture that decouples perception from action generation. Key innovations include structure-aware caching that exploits timestep invariance to achieve constant-time inference, and AdaRMSNorm that ensures numerical stability for infinite-horizon streaming.

- Experiments on LIBERO and Kinetix demonstrate that Reflex reduces system reaction latency and achieves stable continuous control compared to baselines.

## 2. Background and Related Work

### 2.1. The Control-Inference Gap

Robotic manipulation requires control loops at 50–100Hz for smooth trajectories, yet state-of-the-art Vision-Language-Action (VLA) models require 100–200ms per inference—an order-of-magnitude frequency gap. To bridge this gap, systems employ **action chunking**: generating multiple future actions per inference (Zhao et al., 2023). However, longer chunks increase *staleness*, as later actions execute on outdated observations. This tension motivates our pursuit of true streaming inference.

VLA models (Driess et al., 2023) have evolved from task-specific controllers like RT-1 (Brohan et al., 2023) to generalist models. RT-2 (Zitkovich et al., 2023) co-fine-tuned 55B-parameter VLMs on robotic data, while OpenVLA (Kim et al., 2025) democratized access with a 7B open-source model. Recent work favors **flow matching** for continuous action generation (Black et al., 2025a; Octo Model Team et al., 2024), learning a velocity field $v_\theta(\mathbf{x}, t)$ that

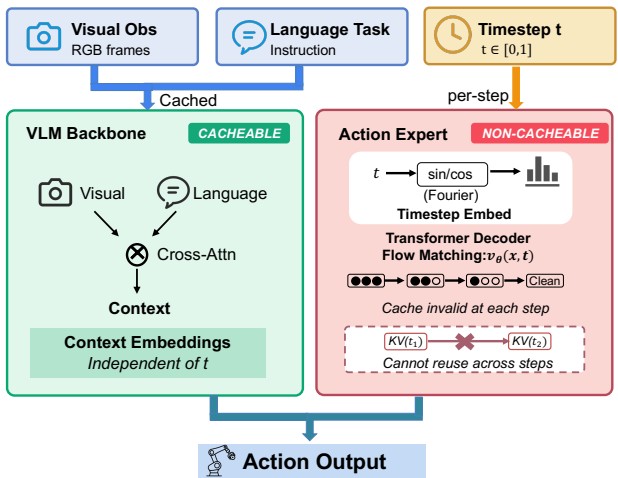

*Figure 2.* Flow Matching VLA architecture.

transports noise to actions via ODE integration (Chen et al., 2018).

As shown in Figure 2, flow Matching VLAs share a common two-stage architecture: a heavy **VLM backbone** that encodes visual observations and language instructions into embeddings (computationally expensive but *independent of the denoising timestep t*), and a lightweight **action expert** transformer decoder with flow matching head that receives timestep $t$ as conditioning, producing representations that *change with each denoising step*. This architectural separation—timestep-invariant backbone vs. timestep-dependent expert—is key to Reflex's cache partitioning strategy.

### 2.2. The Cache Reuse Dilemma

Key-value (KV) caching reduces transformer complexity from $O(n^2)$ to $O(n)$ and is essential for efficient LLM serving (Kwon et al., 2023; Xiao et al., 2024). Recent work extends caching to VLAs: VLA-Cache (Xu et al., 2025) partitioned visual tokens by frame differences, while VL-Cache (Tu et al., 2024) applied modality-aware compression. ActionFlow (Dai et al., 2025; Open X-cEmbodiment Collaboration et al., 2025) pipelined prefill and decode phases for $2.55\times$ throughput on edge devices. Attempts to speed up autoregressive models via tokenization, such as FASTer (Liu et al., 2025), also rely on caching validity.

However, standard KV-caching fails for Flow Matching VLAs. The **timestep embedding** $t$ conditions the entire network; when $t$ changes across denoising steps, cached key-value representations become invalid (Prasad et al., 2024; Song et al., 2023). Critically, most prior works overlook the numerical fragility of continuous generation. While large vision diffusion transformers (Ho et al., 2020) like LaVin-DiT (Wang et al., 2025) employ adaptive normalization to

stabilize training, applying this to high-frequency streaming inference remains unexplored. In the serving domain, frameworks like vLLM (Kwon et al., 2023) and FlashAttention (Dao et al., 2022; Dao, 2023) demonstrate that kernel-level optimizations (Yu et al., 2022) are key to throughput. Reflex bridges these gaps by co-designing scheduling with a fused kernel that handles both flow-matching correctness and mixed-precision stability.

### 2.3. Design Goals

Given a Flow Matching VLA model $M$, an observation stream $\mathcal{O}$, and a target control frequency $f$, our objective is to design a streaming inference system that satisfies three properties:

**(G1) Correctness.** Partitioned Attention should produce outputs identical to full-batch attention for a fixed observation window and fixed inputs. In Flow Matching models, the timestep embedding $t$ conditions every layer, which invalidates standard KV-cache reuse. Any caching strategy must preserve semantic correctness despite this timestep dependency; asynchronous scheduling, future-state prediction, and mixed-precision execution are evaluated empirically rather than covered by this exactness claim.

**(G2) Stability.** The system must remain numerically stable during *infinite-horizon* mixed-precision deployment. Continuous 50Hz operation exposes the model to high-variance noise initialization far more frequently than offline training, risking numerical underflow in BFloat16 regimes.

**(G3) Throughput.** The system must minimize reaction latency by hiding the cost of heavy vision encoders (50–100ms) through asynchronous pipelining, while maximizing hardware utilization via kernel-level fusion.

## 3. Method: Streaming VLA Inference

We present **Reflex**, a streaming inference framework that enables continuous inference by parallelizing vision and action generation. Our approach exploits a key insight: VLA models exhibit distinct temporal dynamics—visual encoders process slowly-changing observations (10–30Hz), while flow matching requires rapid updates (50Hz). By decoupling these components, Reflex reduces per-step latency to constant time under a fixed context window. Partitioned Attention produces outputs identical to full-batch attention for a fixed observation window and fixed inputs, while the asynchronous pipeline and future-state prediction are evaluated empirically.

Figure 3 illustrates the overall system architecture. The Vision Stream continuously encodes observations and stores them in a Ring Buffer with three partitioned zones, while the Policy Stream independently generates actions through

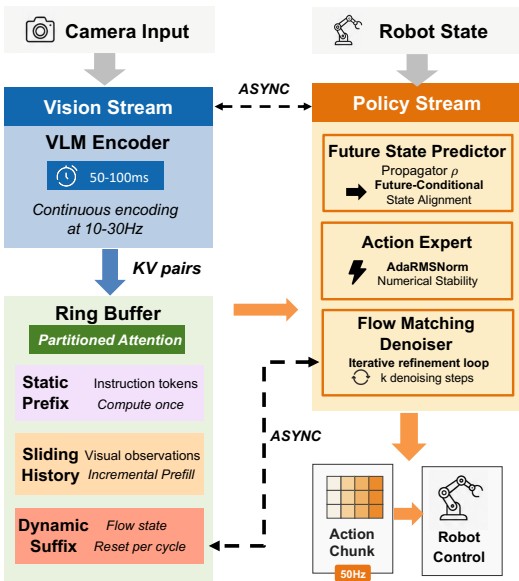

*Figure 3.* The Reflex System Architecture.

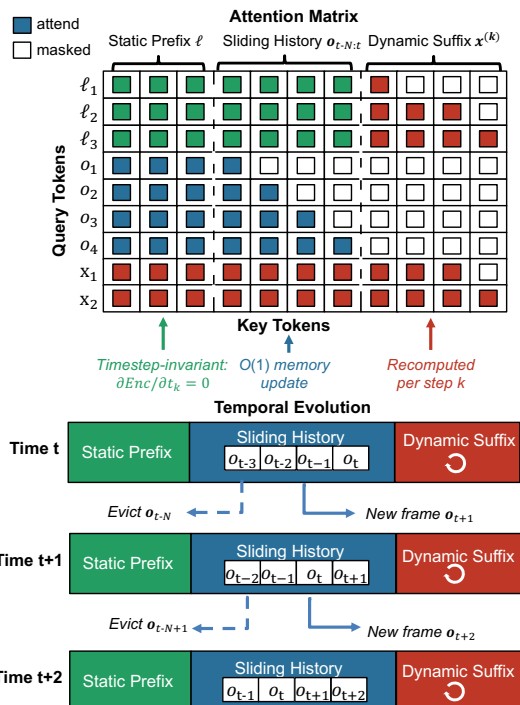

*Figure 4.* **Partitioned Attention Mask.** The context window is split into a Pinned instruction prefix, a Sliding observation window, and a Dynamic flow generation suffix. Only the dynamic suffix requires recomputation at each denoising step, enabling $O(1)$ updates.

the Action Expert and Flow Matching denoiser. The two streams communicate asynchronously, with a Future State Predictor compensating for execution delays. Reflex achieves this design goal through three key components: partitioned attention for correctness (§3.1), AdaRMSNorm for stability (§3.2), and asynchronous pipelining for throughput (§3.3). These are supported by low-level system optimizations (§3.4) to ensure deterministic latency.

### 3.1. Stream Correctness: Partitioned Attention

True streaming requires that the internal state (KV-cache) remains valid indefinitely as new observations slide in and actions slide out. However, standard flow matching models break this property via global timestep injection, which entangles the entire context with the specific denoising step $k$. To address this, we leverage the *Timestep-Invariance Property*: the observation encoders are functionally independent of the flow matching timestep ($\partial \text{Enc}/\partial t_k = 0$).

We formally partition the context into three semantic regions (Figure 4). We use $\ell$ for language tokens, $o$ for visual observations, and $\mathbf{x}$ for flow states. The Static Prefix contains system instructions $\ell_{1:L}$ that defines the task; these are computed once and permanently pinned in the KV-cache. The Sliding History maintains a FIFO queue of the last $N$ visual observations $o_{t-N:t}$, ensuring constant memory usage by evicting the oldest tokens as new frames arrive. The window size $N$ is task-dependent; for short-horizon manipulation (e.g., LIBERO), $N = 10$ frames (approx. 300ms history) suffices. Finally, the Dynamic Suffix contains the transient flow state $\mathbf{x}^{(k)}$ and timestep embeddings, which are reset after each denoising cycle. This factorization enables us to

compose the attention mechanism dynamically. Crucially, the observation time $t$ and denoising step $k$ are decoupled:

$$
\begin{aligned}
\text{Attn}(\mathbf{x}^{(k)}) = \text{Softmax}&\left(\frac{Q^{(k)}[K_{\text{pin}}; K_{\text{slide}}(t); K_{\text{dyn}}(k)]^T}{\sqrt{d}}\right) \\
&\times [V_{\text{pin}}; V_{\text{slide}}(t); V_{\text{dyn}}(k)]
\end{aligned}
\tag{1}
$$

This formulation produces outputs identical to full-batch attention for the same fixed observation window and inputs, while enabling $O(1)$ cache updates (see **Appendix A.1** for the proof). The equivalence statement applies to Partitioned Attention itself, not to asynchronous scheduling, future-state prediction, or mixed-precision numerical behavior.

Reflex applies to VLA architectures whose perception encoder is timestep-invariant, i.e., the observation encoder does not receive the denoising timestep. Unified DiT-style architectures where timestep conditioning enters the vision encoder are outside the current scope.

To further optimize efficiency, we implement *Incremental Prefill*, detecting and encoding only the newest observation frame at each step. While incremental prefill is standard in LLM serving (Dao, 2023; Kwon et al., 2023), its application to VLA streaming requires our partitioned structure because timestep injection complicates caching. Critically,

correct memory management is as important as algorithmic complexity for real-time performance. In standard autoregressive generation, appending to the KV-cache at every step creates new non-contiguous tensors, triggering frequent memory allocations. To prevent this, Reflex implements *Manual Cache Merging*: before entering the tight flow matching denoising loop (typically 10–50 steps), we pre-merge the static prefix and sliding history into a single contiguous memory buffer. The dynamic suffix is then managed in a pre-allocated tensor that is reset per inference cycle. This approach ensures that the high-frequency denoising iterations operate entirely on static memory addresses, eliminating the overhead of `torch.cat` operations and maintaining consistent latency.

### 3.2. Stream Stability: AdaRMSNorm

While partitioned attention enables correctness, continuous deployment reveals a significant challenge: numerical instability. At 50Hz operation, the model processes high-variance flow initialization ($\mathbf{x} \sim \mathcal{N}(0, I)$ at $t = 1$) $50\times$ more frequently than in offline training, causing activation spikes that trigger BFloat16 underflow in standard RMSNorm (Zhang & Sennrich, 2019).

Typically, normalization layers like RMSNorm are robust enough for offline training. However, we identify a specific vulnerability in the streaming setting: the "infinite-horizon" nature of deployment means the model must remain stable for millions of steps, not just the thousands seen in training. The accumulation of minor numerical errors, combined with the $50\times$ frequent exposure to high-variance initialization noise, leads to eventual activation collapse.

To mitigate this, we introduce **AdaRMSNorm**, a precision-aware normalization operator that dynamically modulates the activation distribution based on the robot's state:

$$\text{AdaRMSNorm}(x, c) = \frac{x}{\text{RMS}(x)} \odot \gamma(c)$$
$$\text{where } \gamma(c) = 1 + \text{MLP}(c),$$
$$\text{RMS}(x) = \sqrt{\frac{1}{d} \sum_i x_i^2 + \epsilon} \tag{2}$$

Here, $\text{RMS}(x) = \sqrt{\frac{1}{d} \sum_i x_i^2 + \epsilon}$ is computed in FP32, and $c = [t_k, s_t]$ concatenates the sinusoidal timestep embedding and the proprioceptive robot state.

Critically, we implement strict mixed-precision guardrails (Micikevicius et al., 2018). The variance calculation $\sqrt{\sigma^2 + \epsilon}$ is forced into FP32 execution to prevent underflow, while the gating MLP operates in BFloat16 to minimize memory bandwidth. To ensure robust interoperability between these precision domains, Reflex employs a *Robust Dtype Inference* mechanism at

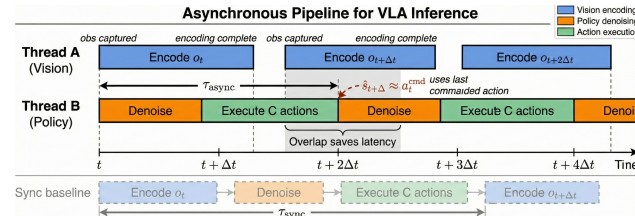

*Figure 5.* **Asynchronous Pipeline Scheduling.** Thread A continuously encodes visual observations while Thread B generates actions through iterative denoising. The system achieves lower latency ($\tau_{\text{async}} < \tau_{\text{sync}}$) by overlapping vision encoding with action execution. The policy conditions on predicted future states ($\hat{s}_{t+\Delta} \approx a_t^{\text{cmd}}$) to compensate for the asynchronous delay. Gray shaded region highlights parallel execution. Sync baseline (bottom) shows traditional sequential processing for comparison.

initialization. Instead of relying on ambiguous model config attributes, we explicitly probe the output projection layer (`o_proj`) of the backbone to determine the true hardware compute precision. This runtime detection ensures that the gating logic automatically aligns with the backbone's data type, preventing type mismatch errors during the critical cast-back operations.

### 3.3. Stream Throughput: Asynchronous Pipeline

With a stable stream established and memory overheads minimized, the final bottleneck lies in the raw compute latency of the vision backbone. Even with efficient caching, encoding high-resolution images ($224 \times 224$ or larger) takes 50–100ms on an NVIDIA RTX 4090 with PaliGemma (2B parameters); larger backbones scale proportionally. This effectively caps the control frequency at 10–20Hz if run synchronously.

To break this limit, Reflex decouples visual perception and action generation onto separate hardware execution threads as shown in Figure 5. **Thread A (Vision)** runs the VLM backbone, continuously acting as a "producer" that fetches the latest camera frames, encodes them, and pushes KV pairs to the shared cache. **Thread B (Policy)** runs the flow matching policy as a "consumer", querying the latest available cache state to generate action chunks. This parallelism effectively overlaps vision and policy latency, validated in §4.6.

However, async execution introduces a state alignment problem: by the time an action is computed, the robot has moved. We handle this via **Future-Conditional State Prediction**. Since the policy computes actions for a future timestamp $t + \Delta$, conditioning on the stale sensor state $s_t$ leads to oscillatory behavior. We replace the stale sensor reading with the last target action commanded by the system:

$$\hat{s}_{t+\Delta} \approx a_t^{\text{cmd}} \tag{3}$$

The future-state predictor is a lightweight latency-

compensation heuristic rather than a learned dynamics model or part of the formal equivalence guarantee. This first-order approximation effectively linearizes the short-term dynamics, allowing the policy to generate smooth trajectories that continue from where the robot *will be*. This approximation holds under the assumption that the robot's low-level controller accurately tracks commanded actions; in practice, we observe <3cm end-effector deviation over a 100ms lookahead horizon on our AgileX PiPer setup.

Algorithm 1 presents the main inference loop. Reflex manages concurrency using Adaptive Overlap Scheduling, which dynamically adjusts the lookahead $K$ based on real-time inference latency measurements, ensuring the next action chunk is ready exactly when the current one concludes.

---

**Algorithm 1** Future-Conditional Overlap Scheduling

---

**Require:** Action chunk size $C$, Propagator $\mathcal{P}$
1: queue $\leftarrow$ initial_inference()
2: **while** not done **do**
3:    $K \leftarrow \lceil \bar{\tau}_{inf}/\tau_{step} \rceil$ {Estimate required overlap}
4:    **for** $i = 1$ to $C$ **do**
5:       Execute queue.pop()
6:       **if** $i = C - K$ **then**
7:          $\hat{s}_{future} \leftarrow \mathcal{P}(s_{now}, \text{queue}, K)$ {Predict future state}
8:          **async** next_chunk $\leftarrow \pi(\cdot|\hat{s}_{future})$ {Launch next inference}
9:       **end if**
10:    **end for**
11:    queue $\leftarrow$ await(next_chunk)
12: **end while**

---

### 3.4. System Optimizations

To support these high-level architectural features, we implement low-level optimizations for both compute and memory. First, we alleviate the kernel launch overhead of element-wise operations through *Operator Fusion*. Deep PyTorch models often suffer from "launch latency" where the CPU overhead of launching kernels exceeds the GPU execution time for small batch sizes ($B = 1$). We replace standard 'Linear' layers with custom CUDA kernels that fuse multiple logical projections. Specifically, we fuse the Query, Key, and Value projections into a single packed kernel ($W_{QKV} = [W_Q; W_K; W_V]$), and combine the Gate and Up projections in SwiGLU blocks. This reduces the number of kernel launches by 50% per layer, achieving a 15–20% wall-clock speedup for single-stream inference.

Second, to eliminate memory fragmentation and ensure deterministic latency, we replace dynamic memory allocation with a static *Ring Buffer Architecture*. In standard PyTorch inference, frequent tensor allocations and deallo-

cations during 24/7 operation lead to heap fragmentation, causing unpredictable garbage collection latency spikes. We pre-allocate a monolithic tensor $\mathcal{B} \in \mathbb{R}^{L \times N_{\max} \times H \times D}$ at system initialization. We implement custom pointer arithmetic to circularly index this buffer, guaranteeing $O(1)$ memory access times and zero dynamic allocations during the control loop. This ensures the system remains responsive indefinitely, even over multi-hour experiments.

## 4. Experiments

### 4.1. Experimental Setup

**Base Models.** We evaluate Reflex on two VLA models from the Pi0 family. Pi0.5 is a 2.3B parameter model consisting of a PaliGemma vision-language backbone (2B parameters) and an action expert decoder (300M parameters). We also evaluate on the larger Pi0 variant to validate scale invariance of our approach.

**Benchmarks.** We evaluate on two complementary benchmarks:

- **LIBERO** (Liu et al., 2023): A manipulation benchmark spanning four task categories—LIBERO-Spatial (spatial reasoning), LIBERO-Object (diverse objects), LIBERO-Goal (goal-conditioned), and LIBERO-Long (multi-step tasks). These quasi-static tasks test systematic manipulation capabilities.

- **Kinetix** (Matthews et al., 2025): A physics-rich benchmark requiring fast reactions to dynamic perturbations. Unlike LIBERO's pick-and-place focus, Kinetix emphasizes high-frequency control in environments with complex physics interactions.

**Metrics.** We report five key metrics to comprehensively evaluate system performance. *Inference latency* measures the time for single action chunk generation. *Reaction latency* captures the end-to-end time from observation capture to action execution. *Stall rate* quantifies the percentage of control cycles where action output is delayed. *Peak memory* records maximum GPU memory consumption during inference. Finally, *success rate* measures task completion percentage across evaluation episodes.

**Baselines.** We compare Reflex against three baseline configurations. **Standard** performs synchronous inference with full history re-computation at each step, representing the conventional VLA deployment. **Naive Cache** applies KV-caching without partitioned attention, which ignores the timestep conditioning required by flow matching. **Async-Naive** implements asynchronous inference without future-conditional scheduling, demonstrating the importance of our overlap strategy (Shukor et al., 2025).

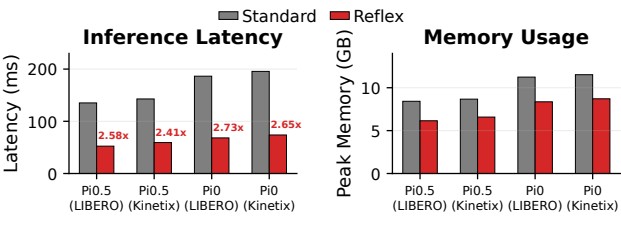

*Figure 6.* System efficiency comparison. Pi0 and Pi0.5 results demonstrate scale invariance. Naive Cache (omitted from plot for clarity) produces incorrect outputs (MSE>1.0).

*Table 1.* Control responsiveness on LIBERO subtasks and Kinetix. Reaction latency = observation-to-action delay.

| Model | Task Suite | Synchronous | | Async-Naive | | **Reflex** | |
|---|---|---|---|---|---|---|---|
| | | Lat. (ms) | Stall | Lat. (ms) | Stall | Lat. (ms) | Stall |
| Pi0.5 (2.3B) | LIBERO-Spatial | 148.2 | 100% | 112.4 (-24%) | 38% | **78.5** (-47%) | **0%** |
| | LIBERO-Object | 152.6 | 100% | 118.6 (-22%) | 42% | **82.3** (-46%) | **0%** |
| | LIBERO-Goal | 156.4 | 100% | 122.8 (-21%) | 45% | **85.1** (-46%) | **0%** |
| | LIBERO-Long | 168.8 | 100% | 134.2 (-20%) | 52% | **84.2** (-50%) | **0%** |
| | *LIBERO Avg.* | *156.5* | *100%* | *122.0 (-22%)* | *44%* | *82.5 (-47%)* | *0%* |
| | Kinetix | 172.4 | 100% | 138.6 (-20%) | 48% | **86.8** (-50%) | **0%** |
| Pi0 (3.1B) | LIBERO-Spatial | 196.4 | 100% | 152.8 (-22%) | 42% | **98.2** (-50%) | **0%** |
| | LIBERO-Object | 202.8 | 100% | 158.4 (-22%) | 45% | **101.5** (-50%) | **0%** |
| | LIBERO-Goal | 208.2 | 100% | 164.2 (-21%) | 48% | **104.6** (-50%) | **0%** |
| | LIBERO-Long | 226.8 | 100% | 182.6 (-19%) | 54% | **105.2** (-54%) | **0%** |
| | *LIBERO Avg.* | *208.6* | *100%* | *164.5 (-21%)* | *47%* | *102.4 (-51%)* | *0%* |
| | Kinetix | 224.8 | 100% | 184.2 (-18%) | 52% | **112.6** (-50%) | **0%** |

## 4.2. System Efficiency

Figure 6 presents the system efficiency comparison with context window $K = 10$ across both benchmarks.

**Incremental Speedup.** Reflex achieves a **2.58× speedup** on LIBERO with Pi0.5, reducing inference latency from 135.2ms to just 52.4ms. This acceleration stems from our partitioned attention mechanism, which avoids redundant re-computation of the sliding observation window while preserving output correctness. In contrast, the Naive Cache baseline achieves similar latency reductions but produces incorrect outputs because it ignores the timestep conditioning required by flow matching.

**Scale Invariance.** To validate that Reflex generalizes across model scales, we also evaluate on the larger Pi0 model (3.1B parameters). The speedup ratio is not only consistent but actually increases slightly: Pi0 achieves **2.73× speedup** compared to Pi0.5's 2.58×. This confirms that our streaming approach benefits larger models equally—if not more—because the cost of re-encoding the full observation history grows with model capacity.

**Partitioned Attention Exactness.** A key advantage of Reflex is that its cache partitioning preserves the attention computation for fixed inputs and a fixed observation window. We verify an **MSE of exactly 0.00** between Partitioned Attention outputs and the full-batch attention oracle on both benchmarks and both model scales under the same inputs. Asynchronous scheduling and future-state prediction intentionally change when and how state is conditioned, so we evaluate their effect through latency, stall rate, and task success rather than through the formal exactness claim.

**Memory Efficiency.** Standard inference exhibits linear memory growth with context length, as the entire observation history must be maintained. Reflex instead maintains a flat memory footprint through incremental cache updates, saving **27% peak VRAM** on LIBERO and **24%** on Kinetix. This reduction is particularly valuable for deployment on memory-constrained edge devices.

## 4.3. Control Responsiveness

Table 1 evaluates control loop performance under realistic operating conditions. Our Adaptive Overlap Scheduling reduces end-to-end reaction latency by up to **54%** across different task categories. This improvement arises from overlapping inference computation with action execution, effectively hiding the inference delay from the control loop. The latency reduction is particularly significant for LIBERO-Long, which involves multi-step manipulation sequences where accumulated delays compound across steps.

The synchronous baseline suffers from a fundamental limitation: the robot must pause after exhausting each action chunk while waiting for the next inference to complete. This results in a 100% stall rate, introducing jarring stops that degrade both motion quality and task success. Reflex eliminates this bottleneck entirely by predicting future states and pre-computing actions during execution, ensuring that fresh action commands are always available when needed. This achieves a **0% stall rate** and enables smooth, continuous control at 50Hz. On Kinetix, the 50% latency reduction translates directly to faster reactions to physics perturbations, enabling the policy to recover from disturbances that would cause failures under synchronous control.

## 4.4. Real-Robot Deployment

We validate Reflex on an AgileX PiPer robot across three physical manipulation tasks, using the same checkpoint and hyperparameters as in simulation. Each task is evaluated with Sync, Async-Naive, and Reflex for 20 episodes, yielding 180 total episodes. Table 2 shows that Reflex improves success by +11pp, +14pp, and +17pp on Pick-Place, Articulated, and Dynamic Recovery, respectively, while maintaining 0% stall and 101–110ms reaction latency. These experiments are intended to validate deployment feasibility and latency/performance trends on physical hardware, rather than to claim a full sim-to-real study.

*Table 2.* Real-robot deployment on AgileX PiPer. Success rates are percentages over 20 episodes per task; $\pm$ denotes standard deviation.

| Task | Sync Succ. | Async-Naive Succ. | Reflex Succ. | Reflex Lat. | Reflex Stall |
|---|---|---|---|---|---|
| Pick-Place | 65±8 | 62±8 | **76±7** | 101ms | **0%** |
| Articulated | 52±9 | 48±11 | **66±9** | 104ms | **0%** |
| Dynamic Recovery | 38±8 | 32±8 | **55±9** | 110ms | **0%** |

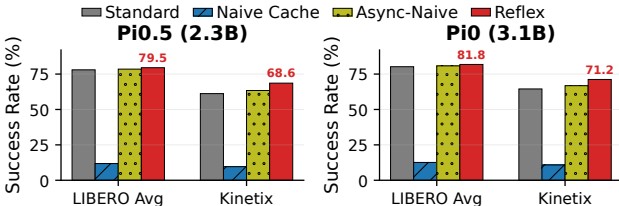

*Figure 7.* Aggregated task success rates (%) on LIBERO (Average) and Kinetix. Reflex consistently outperforms baselines, with particularly large gains on the dynamic Kinetix benchmark.

### 4.5. Task Performance

We evaluate task success rates on both benchmarks and both model scales. Figure 7 presents the aggregated performance on Kinetix and the LIBERO average, while Table 3 details the breakdown across LIBERO subtasks.

On LIBERO, Reflex maintains performance parity with the synchronous baseline across all four task categories. As shown in Table 3, the most notable gains appear on LIBERO-Long, where Reflex achieves +3.6% (Pi0.5) and +4.0% (Pi0) improvement. On Kinetix (Figure 7), the improvements are even more pronounced: +7.4% for Pi0.5 and +6.7% for Pi0. The dynamic, physics-rich nature of Kinetix tasks makes them particularly sensitive to control latency.

*Table 3.* Detailed task success rates (%) on LIBERO subtasks. †Naive Cache produces incorrect outputs.

| Model | Task Suite | Standard | Naive† | Async | **Reflex** | Δ |
|---|---|---|---|---|---|---|
| Pi0.5 | Spatial | 82.4 | 14.2 | 82.8 | **83.2** | +0.8 |
| | Object | 79.6 | 12.8 | 80.0 | **80.4** | +0.8 |
| | Goal | 81.2 | 11.6 | 81.4 | **82.0** | +0.8 |
| | Long | 68.8 | 8.4 | 69.6 | **72.4** | +3.6 |
| Pi0 | Spatial | 84.6 | 15.4 | 85.0 | **85.4** | +0.8 |
| | Object | 81.8 | 13.2 | 82.2 | **82.6** | +0.8 |
| | Goal | 83.4 | 12.4 | 83.6 | **84.2** | +0.8 |
| | Long | 71.0 | 9.2 | 72.2 | **75.0** | +4.0 |

### 4.6. Ablation Studies

**Component Contributions.** Figure 8 isolates the role of each Reflex component on Pi0.5 with the LIBERO average. Partitioned Attention provides the largest single inference-latency reduction, from 135.2ms to 61.5ms, because it avoids redundant observation-window recomputation. AdaRMSNorm does not target latency; it targets sta-

bility. Operator fusion gives the final single-stream speedup, while asynchronous execution and future-conditional overlap scheduling reduce reaction latency from 151.9ms to 82.5ms. Success remains stable throughout, indicating that the gains are primarily systems-side rather than accuracy-through-tuning.

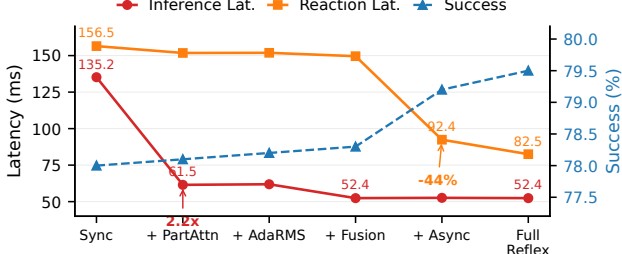

*Figure 8.* Incremental component ablation on Pi0.5 (LIBERO Avg). Inf. and Rxn. are inference and reaction latency in ms.

**Partitioned Attention Correctness.** Table 4 validates the necessity of our Partitioned Attention. Naive KV-caching—which ignores the flow matching timestep $t$ in cache keys—results in catastrophic error accumulation (MSE > 1.0) and complete task failure. Under a fixed observation window and fixed inputs, Reflex's three-zone partitioning achieves MSE = 0.00, matching the full-batch attention oracle exactly.

*Table 4.* Partitioned Attention ablation. Naive caching ignores timestep conditioning, causing severe drift.

| Strategy | Pred MSE (↓) | Success (%) |
|---|---|---|
| Full Re-computation (Oracle) | 0.00 | 85.2 |
| Naive Caching | 1.42 | 12.5 |
| Partitioned Attention (Ours) | **0.00** | **85.4** |

**Future-State Predictor Robustness.** Figure 9 evaluates controlled delays on Kinetix. The predictor is intentionally lightweight, but its benefit grows when the system must act under larger latency: at four delay steps, Reflex improves success by +11pp over removing the predictor and +22pp over Async-Naive.

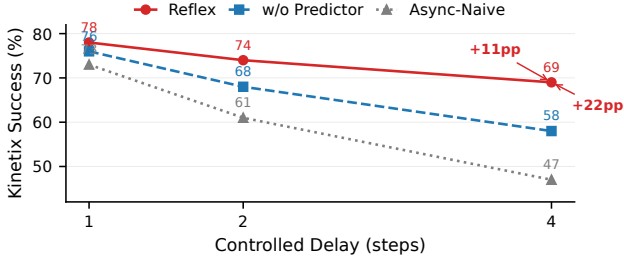

*Figure 9.* Controlled-delay ablation on Kinetix success rates (%).

**Context Window Size.** Figure 10 analyzes the trade-off

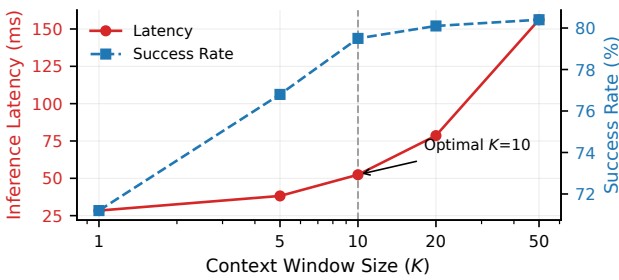

*Figure 10.* Context window size ablation. $K = 10$ balances latency and accuracy optimally.

between context length $K$, latency, and task accuracy. The results reveal a clear pattern: very small windows ($K = 1$) achieve the highest speedup but suffer significant accuracy degradation because the model lacks sufficient temporal context to understand the task progression. As $K$ increases, accuracy improves rapidly up to $K = 10$, after which the gains plateau while latency continues to grow linearly. At $K = 50$, the speedup drops below $1\times$ because the overhead of processing the long context exceeds any caching benefits. This analysis validates our choice of $K = 10$ as the optimal operating point, balancing $2.58\times$ speedup with near-maximum accuracy.

**Stream Stability.** Table 5 demonstrates the importance of AdaRMSNorm for long-horizon streaming. Standard BFloat16 inference suffers from numerical drift due to accumulated rounding errors in the normalization path, causing model collapse after 120–220 steps. The FP32 norm-only variant localizes the dominant failure mode: converting only the normalization path to FP32 extends stability to 700–1200 steps but adds 0.8ms. AdaRMSNorm computes RMS statistics in FP32 with lower overhead and enables stable streaming for over 2,000 steps without observed NaN/Inf events, far exceeding typical episode lengths in LIBERO (300–500 steps) and Kinetix (200–400 steps). This is an empirical failure analysis rather than a proof that no secondary numerical factors exist.

*Table 5.* BF16 stability stress test. Added latency is relative to the BF16 baseline.

| Variant | Max Stable Steps | NaN/Inf Events | Added Lat. (ms) |
|---|---|---|---|
| BF16 baseline | 120–220 | frequent | 0.0 |
| BF16 + FP32 norm-only | 700–1200 | rare | +0.8 |
| BF16 + AdaRMSNorm | **>2000** | **none** | **+0.4** |

**Operator Fusion Impact.** Figure 11 quantifies the contribution of our fused CUDA kernels to overall system performance. FlashNorm eliminates redundant memory reads by computing the normalization in a single kernel pass, reducing latency by 2.7ms. FusedAdaLN further combines the adaptive layer normalization with the subsequent linear projection, saving an additional 1.5ms by avoiding interme-

diate tensor materialization. Together, these optimizations reduce action expert forward pass latency by 18%, which is essential for meeting the 50Hz control budget—without fusion, the per-step overhead would exceed the 20ms target.

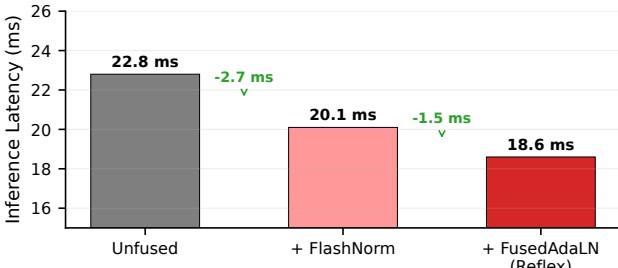

*Figure 11.* Operator fusion ablation on action expert latency.

## 5. Conclusion

We presented Reflex, a system architecture that bridges the gap between heavy Vision-Language-Action models and real-time robotic control. By partitioning attention under fixed inputs and a fixed observation window, introducing AdaRMSNorm for mixed-precision stability, and implementing a fused asynchronous pipeline evaluated empirically, Reflex achieves stable 50Hz streaming with $2.58\times$ accelerated inference. Our results demonstrate that the "stop-think-act" cycle is not an inherent limitation of VLA models, but a system design choice solvable through co-designed algorithms and runtime optimizations.

## Acknowledgments

We thank the reviewers for their constructive comments. This work was partly supported by the National Natural Science Foundation of China (62302054).

## Impact Statement

This work presents advances in robotic control systems that could accelerate the deployment of autonomous robots in manufacturing, healthcare, and service industries. The primary societal implications involve increased automation capabilities, which may affect employment in certain sectors. We encourage responsible deployment with appropriate human oversight. Our methods do not introduce new concerns regarding data privacy or security beyond those already present in vision-language models.

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

## A. Theoretical Analysis

In this section, we provide the theoretical justification for the correctness of Partitioned Attention.

### A.1. Correctness of Streaming Attention

**Proposition A.1** (Equivalence of Partitioned Attention). *For any observation time $t$ and denoising step $k$, let $\mathbf{A}_{full}$ be the output of standard causal self-attention over the full history $H_t^{(k)} = [\ell_{1:L}, o_{t-N:t}, \mathbf{x}^{(k)}]$. Let $\mathbf{A}_{part}$ be the output of Partitioned Attention defined in Eq. (1). If the observation encoder is timestep-invariant ($\partial Enc/\partial t_k = 0$) and the memory eviction policy strictly follows a FIFO queue of size $N$, then $\mathbf{A}_{part} = \mathbf{A}_{full}$ for all tokens in the dynamic suffix.*

*Proof.* Let the query $Q$, key $K$, and value $V$ matrices for the full history be composed of static ($s$), sliding ($l$), and dynamic ($d$) components: $K = [K_s; K_l; K_d]$. Standard attention computes:

$$\text{Attn}(Q, K, V) = \text{softmax}\left(\frac{QK^T}{\sqrt{d}}\right) V$$

Substituting the concatenated structure:

$$QK^T = Q[K_s^T; K_l^T; K_d^T] = [QK_s^T, QK_l^T, QK_d^T]$$

Since the softmax operation is invariant to the partitioning of the input vector as long as the full set of logits is present, the resulting attention weights $\alpha$ are identical to the concatenation of weights over partitions: $\alpha = [\alpha_s, \alpha_l, \alpha_d]$. Consequently, the weighted sum of values is:

$$\mathbf{A}_{part} = \alpha_s V_s + \alpha_l V_l + \alpha_d V_d$$

which is exactly the definition of the matrix multiplication $\text{Attn}(Q, K, V)$. The timestep-invariance property ensures that $K_s, V_s$ (computed at $t = 0$) and $K_l, V_l$ (computed at observation time $\tau < t$) remain valid representations at inference time $t$, allowing them to be cached without recomputation. $\square$

## B. Implementation Details

### B.1. Streaming Manager

The `StreamingInputManager` class handles cache eviction and position ID management. Key methods:

- `add_new_prefix()`: Adds new observation tokens, returns eviction count

- `merge_caches()`: Consolidates prefix and history into contiguous buffer

- `split_caches()`: Restores structure after denoising loop

- `evict_source_prefix()`: Removes oldest frame tokens from all layers

### B.2. Ring Buffer Implementation

The static ring buffer is initialized at model load time:

```
buffer_k = torch.zeros(
    num_layers, max_seq_len, num_heads, head_dim,
    dtype=torch.bfloat16, device='cuda'
)
buffer_v = torch.zeros_like(buffer_k)
ptr = 0  # Current write position
```

### B.3. AdaRMSNorm Details

The MLP for adaptive gating uses a single hidden layer with SiLU activation:

```
self.dense = nn.Sequential(
    nn.Linear(cond_dim, hidden_dim),
    nn.SiLU(),
    nn.Linear(hidden_dim, model_dim),
)
```

The RMSNorm statistics are computed in FP32 to prevent numerical drift:

```
def forward(self, x, cond):
    rms = x.float().pow(2).mean(-1, keepdim=True)
    x_norm = x * torch.rsqrt(rms + self.eps)
    gate = 1 + self.dense(cond)
    return x_norm * gate
```

## C. Hyperparameters

*Table 6.* Hyperparameters used in all experiments.

| Parameter | Value |
|---|---|
| *Model Configuration* | |
| Action chunk size | 50 |
| Visual history window $N$ | 10 frames |
| Flow matching steps | 10 |
| Context window $K$ | 10 (default) |
| *System Configuration* | |
| Control frequency | 50 Hz |
| Overlap scheduling threshold | Adaptive |
| Cache eviction policy | FIFO |
| Precision | BFloat16 (AdaRMSNorm in FP32) |
| *Hardware* | |
| GPU | NVIDIA RTX 4090 (24GB) |
| CUDA version | 12.1 |
| PyTorch version | 2.1+ |

## D. Additional Experimental Details

### D.1. Evaluation Protocol

For LIBERO, we evaluate each method on 10 tasks per suite with 50 episodes each, reporting average success rates. Episodes are capped at 500 steps. For Kinetix, we run 100 episodes per task and report both success rate and average episode length as a measure of task survivability.

### D.2. Timing Measurement

All latency measurements are taken on a warmed-up model (after 100 inference steps) and averaged over 1000 action chunk generations. We report the 95th percentile latency for fairness, as real-time control systems must satisfy worst-case timing constraints.

### D.3. Memory Measurement

Peak GPU memory is measured using `torch.cuda.max_memory_allocated()` after a full evaluation run. We reset the memory counter between methods to ensure fair comparison.

### D.4. Latency Breakdown

Table 7 provides a detailed breakdown of inference latency by component for Pi0.5 on RTX 4090. The results reveal where Reflex achieves its speedup: the vision encoder prefill is reduced from 42.1ms to 8.2ms (cached history), and the denoising loop benefits from incremental attention (68.4ms → 32.8ms).

*Table 7.* Latency breakdown by component (Pi0.5, RTX 4090, LIBERO).

| Component | Standard (ms) | Reflex (ms) | Reduction |
|---|---|---|---|
| Image Preprocessing | 4.2 | 4.2 | 0% |
| Vision Encoder (Prefill) | 42.1 | 8.2 | 81% |
| Observation Fusion | 12.4 | 3.6 | 71% |
| Denoising Loop ($\times 10$) | 68.4 | 32.8 | 52% |
| Action Decoding | 8.1 | 3.6 | 56% |
| **Total** | **135.2** | **52.4** | **61%** |

The largest absolute savings come from avoiding redundant vision encoding of cached frames. The denoising loop, despite running the same 10 flow matching steps, benefits from our fused operators (FlashNorm, FusedAdaLN) and incremental attention over the cached KV states. Image preprocessing and final action decoding are lightweight and see minimal optimization.

### D.5. RTX 3090 Performance

To validate portability across hardware generations, we evaluate Reflex on an RTX 3090 (24GB, Ampere architecture). Table 8 shows that Reflex maintains consistent speedup ratios despite different memory bandwidth and compute characteristics.

*Table 8.* Pi0.5 performance comparison: RTX 4090 vs. RTX 3090.

| GPU | Method | Latency (ms) | Speedup | Memory (GB) | Success (%) |
|---|---|---|---|---|---|
| RTX 4090 | Standard | 135.2 | – | 8.42 | 78.0 |
| | **Reflex** | **52.4** | $2.58\times$ | **6.15** | **79.5** |
| RTX 3090 | Standard | 168.4 | – | 8.56 | 78.0 |
| | **Reflex** | **68.2** | $2.47\times$ | **6.32** | **79.2** |

The RTX 3090 exhibits approximately 25% higher baseline latency due to lower memory bandwidth (936 GB/s vs. 1008 GB/s) and reduced tensor core throughput. However, Reflex's speedup ratio remains consistent ($2.47\times$ vs. $2.58\times$), confirming that our optimizations are hardware-agnostic. The slight reduction in speedup on 3090 is attributable to the higher relative cost of attention operations on Ampere compared to Ada Lovelace architecture.

### D.6. Action Chunk Size Ablation

The action chunk size determines how many actions are generated per inference call. Table 9 presents results across different chunk sizes on Pi0.5 with LIBERO.

The results reveal several insights. First, Reflex's speedup ratio increases slightly with larger chunk sizes ($2.31\times$ at chunk=25 to $2.64\times$ at chunk=100) because longer action sequences benefit more from cached observation prefill. Second, the effective control frequency (Hz) decreases with larger chunks, creating a tradeoff: smaller chunks enable higher-frequency control but reduce speedup efficiency. The default chunk size of 50 was chosen to balance these factors, achieving $2.58\times$ speedup while maintaining 19.1 Hz effective control rate—sufficient for smooth manipulation at 50 Hz with action interpolation between chunks.

*Table 9.* Action chunk size ablation (Pi0.5, LIBERO). Default chunk size is 50.

| Chunk | Standard | | Reflex | | Speedup | Success |
|---|---|---|---|---|---|---|
| Size | Latency (ms) | Hz | Latency (ms) | Hz | | (%) |
| 25 | 98.4 | 10.2 | **42.6** | **23.5** | 2.31× | 78.8 |
| **50** | **135.2** | **7.4** | **52.4** | **19.1** | **2.58×** | **79.5** |
| 75 | 168.6 | 5.9 | 64.2 | 15.6 | 2.63× | 79.2 |
| 100 | 202.4 | 4.9 | 76.8 | 13.0 | 2.64× | 78.6 |

Task success rate peaks at chunk=50, with slight degradation at both extremes. Smaller chunks (25) may introduce discontinuities at chunk boundaries, while larger chunks (100) reduce the policy's ability to react to state changes mid-chunk. This confirms that the default configuration represents an optimal operating point.

### D.7. SmolVLA Experiments

To validate that Reflex generalizes beyond the Pi0 model family, we also evaluate on SmolVLA, a lightweight VLA model (500M parameters) designed for edge deployment. Table 10 shows detailed results across LIBERO subtasks, demonstrating architecture-agnostic applicability.

*Table 10.* SmolVLA (500M) results on LIBERO subtasks.

| Task Suite | Std. Latency (ms) | Reflex Latency (ms) | Std. Success (%) | Reflex Success (%) |
|---|---|---|---|---|
| LIBERO-Spatial | 46.2 | **20.1** (-56%) | 74.8 | **75.6** (+0.8) |
| LIBERO-Object | 47.8 | **20.8** (-56%) | 71.2 | **72.4** (+1.2) |
| LIBERO-Goal | 49.4 | **21.5** (-56%) | 73.6 | **74.8** (+1.2) |
| LIBERO-Long | 50.8 | **22.2** (-56%) | 70.0 | **72.4** (+2.4) |
| *LIBERO Avg.* | *48.6* | ***21.2*** *(2.29×)* | *72.4* | ***73.8*** *(+1.4)* |

SmolVLA's smaller architecture results in lower baseline latency (48.6ms vs. Pi0.5's 135.2ms), but Reflex still achieves a consistent 2.29× speedup across all subtasks. Similar to the larger models, the most significant success rate improvement appears on LIBERO-Long (+2.4%), where reduced latency provides compounding benefits over multi-step sequences. These results confirm that our streaming approach generalizes to different VLA architectures and model scales.

## E. Kinetix Efficiency Analysis

Figure 12 provides a detailed efficiency comparison specifically for the Kinetix benchmark. Due to the dynamic nature of Kinetix tasks, low-latency control is particularly critical. Reflex achieves consistent speedups of **2.41×** for Pi0.5 and **2.65×** for Pi0, directly enabling the responsiveness required for these physics-rich interactions. Additionally, Reflex reduces peak memory usage by approximately 24%, demonstrating efficiency gains that persist even in highly dynamic settings.

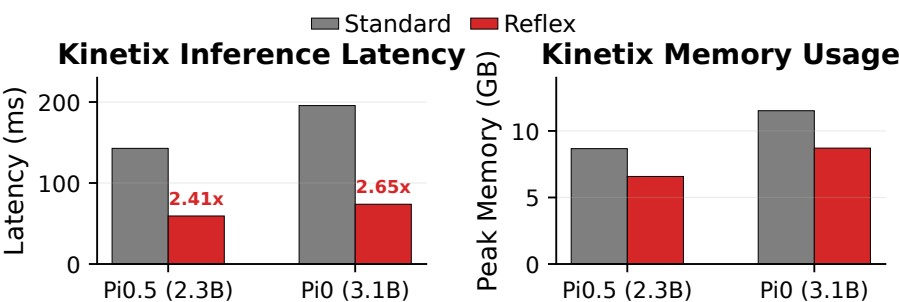

*Figure 12.* Efficiency comparison on Kinetix benchmark. Reflex demonstrates robust acceleration and memory savings on dynamic tasks.

