# OpenReview forum: "Reflex: Real-Time Vision-Language-Action Control  through Streaming Inference"
_ICML.cc/2026/Conference — ICML 2026 regular_

### Official Review · Reviewer_27eJ · 2026-03-11

**Soundness:** 3
**Presentation:** 3
**Significance:** 3
**Originality:** 3
**Overall Recommendation:** 4
**Confidence:** 3

**Summary:**

The paper proposes Reflex, a framework designed to enable real-time, high-frequency control for Vision-Language-Action (VLA) models using flow matching. Standard flow matching is often too slow for dynamic robotics because its iterative denoising process is computationally expensive and incompatible with standard KV-caching. Reflex overcomes these hurdles through three main technical innovations: Partitioned Attention, AdaRMSNorm, and Asynchronous Pipelining. Evaluated on LIBERO and Kinetix benchmarks, Reflex achieves a 2.58x inference speedup, reduces reaction latency by up to 54%, enabling smooth and stable robotic manipulation in dynamic environments.

**Compliance With Llm Reviewing Policy:**

Affirmed.

**Final Justification:**

The real-robot experiments have enhanced the practical value of the work. Additional experiments have addressed my concerns. So I raise my score to 4.

**Key Questions For Authors:**

1. The paper uses AdaRMSNorm to solve bfloat16 instability, but lacks analysis into the root of the instability.
2. Will streaming inference influence policy distribution？especially when sliding window loses historical tokens?
3. Could the approach be validated on real-robot setting?

**Limitations:**

Yes.

**Strengths And Weaknesses:**

Strengths:
1. The partition of attention is interesting. By exploiting "Timestep-Invariance," it only recomputes the dynamic flow state at each denoising step, reducing complexity to O(1) while maintaining mathematical equivalence to full-batch inference.
2. The problem tackled is an important question in VLA, especially for practical deployments.
3. The paper's presentation is good with clear ablations.

Weaknesses:
1. Lack of real-robot experiments, which makes the practical value weakened.

---

> ### Author Rebuttal · Authors · 2026-03-31
>
> We thank the reviewer for the constructive questions. We address each with new experimental evidence.
>
> ## Q1: BF16 Instability Root Cause
>
> Our evidence indicates that the dominant failure mode is a frequency mismatch between training and deployment. At 50Hz streaming, the model encounters high-variance flow initialization ($\mathbf{x} \sim \mathcal{N}(0, I)$ at $t=1$) repeatedly throughout deployment rather than only once per episode as in offline training. Standard BF16 RMSNorm is robust to occasional activation spikes but not their sustained high-frequency occurrence; over long horizons, rounding error in $\sqrt{\sigma^2+\epsilon}$ accumulates and eventually destabilizes the stream. AdaRMSNorm promotes this computation to FP32, stabilizing the normalization path where failures emerge.
>
> New BF16 stress test (E6) provides direct quantification:
>
> | Variant | Max Stable Steps | NaN/Inf Events | Added Latency (ms) |
> |---------|-----------------:|---------------:|--------------------:|
> | BF16 baseline | 120-220 | frequent after collapse | 0.0 |
> | BF16 + FP32 norm-only | 700-1200 | rare | +0.8 |
> | BF16 + AdaRMSNorm | **>2000** | **none observed** | **+0.4** |
>
> The FP32 norm-only ablation localizes the dominant failure mode to the normalization path rather than to BF16 computation in general: converting only this path to FP32 already extends stability from 120-220 steps to 700-1200 steps. AdaRMSNorm then improves both stability and efficiency by computing RMS statistics in FP32 before rescaling, reaching >2000 steps at lower overhead (+0.4ms vs +0.8ms for blanket FP32 normalization). In the revision we will phrase this as an empirical failure analysis rather than a proof that no secondary numerical factors exist.
>
> ## Q2: Sliding Window and Policy Distribution
>
> New K-sweep on **Kinetix** with oracle comparison (E5):
>
> | K | Success (%) | Inference Latency (ms) | Pred MSE to Oracle |
> |---|------------:|-----------------------:|-------------------:|
> | 1 | 72.8 | 38.6 | 0.084 |
> | 5 | 78.3 | 46.9 | 0.021 |
> | 10 | **79.5** | **52.4** | **<1e-6** |
> | 20 | 79.6 | 63.8 | <1e-6 |
> | 50 | 79.4 | 96.2 | <1e-6 |
>
> On the evaluated tasks, outputs and success are effectively unchanged for K≥10, suggesting no practically meaningful distribution shift within this regime. This is consistent with the context-window ablation already reported in the paper. The 0.1pp difference between K=10 and K=20 is within noise; K=10 is preferred for the 17% latency reduction (52.4ms vs 63.8ms). We will revise the text to state this as a benchmark-supported observation, not as a universal guarantee for all tasks or all history lengths.
>
> ## Q3: Real-Robot Validation
>
> We validated on two real-robot platforms. On **AgileX PiPer**, we evaluated 3 tasks × 3 methods × 20 episodes = 180 total episodes using the same checkpoint and hyperparameters as simulation. ± values are standard deviations across episodes.
>
> | Task | Method | Success (%) | Reaction Latency (ms) | Stall (%) |
> |------|--------|------------:|----------------------:|----------:|
> | Pick-Place | Sync | 65±8 | 191 | 100 |
> | Pick-Place | Async-Naive | 62±8 | 148 | 20±8 |
> | Pick-Place | Reflex | **76±7** | **101** | **0** |
> | Articulated | Sync | 52±9 | 198 | 100 |
> | Articulated | Async-Naive | 48±11 | 155 | 32±9 |
> | Articulated | Reflex | **66±9** | **104** | **0** |
> | Dynamic Recovery | Sync | 38±8 | 205 | 100 |
> | Dynamic Recovery | Async-Naive | 32±8 | 160 | 33±9 |
> | Dynamic Recovery | Reflex | **55±9** | **110** | **0** |
>
> Reflex improves success by **+11pp** (Pick-Place), **+14pp** (Articulated), and **+17pp** (Dynamic Recovery) over Sync, with **0% stall** across all tasks.
>
> On **WidowX** with **X-VLA-0.9B**, we ran a real-robot Pick-Place evaluation:
>
> | Platform | Method | Success (%) | Reaction Latency (ms) |
> |----------|--------|------------:|----------------------:|
> | WidowX | Sync | 81 | 112 |
> | WidowX | Reflex | **85** | **48** |
>
> These results confirm that Reflex improves real-robot responsiveness and maintains or improves task success across both platforms.
>
> Importantly, these hardware experiments use the same checkpoints and hyperparameters as the simulation runs; they are intended to validate deployment feasibility and the latency/performance trend on physical systems, not to claim an exhaustive sim-to-real study.

---

> > ### Author Rebuttal · Reviewer_27eJ · 2026-04-03
> >
> > Thanks for the detailed reply. I will consider raising my score to 4.

---

> > > ### Author Response · Authors · 2026-04-07
> > >
> > > Thank you very much for your response and for your careful consideration. We sincerely appreciate your time and feedback.

---

### Official Review · Reviewer_cjJe · 2026-03-12

**Soundness:** 3
**Presentation:** 2
**Significance:** 2
**Originality:** 3
**Overall Recommendation:** 3
**Confidence:** 3

**Summary:**

This paper proposes Reflex, a streaming inference architecture for flow-matching Vision-Language-Action (VLA) models to enable real-time robotic control. The key idea is to decouple perception and action generation through asynchronous execution and a partitioned attention cache, allowing incremental updates while maintaining correctness with respect to full-batch inference. The system further introduces AdaRMSNorm for numerical stability in long-horizon streaming and several low-level optimizations to reduce latency. Experiments on LIBERO and Kinetix demonstrate improved control responsiveness and inference efficiency while maintaining comparable task success rates.

**Compliance With Llm Reviewing Policy:**

Affirmed.

**Final Justification:**

The rebuttal provides clear responses and additional experimental evidence addressing my concerns. I appreciate the clarification of the equivalence claim and the added evaluations on predictor robustness and real-world deployment.

While these responses improve the clarity and reliability of the work, they do not substantially change my assessment of the overall contribution, which remains primarily system-oriented. I will therefore maintain my current score.

**Key Questions For Authors:**

1. The paper repeatedly claims mathematical equivalence to full-batch inference. However, the experimental verification appears limited to the partitioned attention mechanism (Table 3). Does this equivalence hold for the entire streaming pipeline, including asynchronous execution, future-conditional state prediction, and mixed-precision inference?

2. The asynchronous policy stream relies on a future-conditional state predictor to compensate for control latency. However, the paper provides limited analysis of how prediction errors affect downstream policy execution. How sensitive is Reflex to inaccuracies in the predicted future state, especially in dynamic manipulation scenarios where environment changes may violate the short-horizon linear approximation?

3. The proposed partitioned attention relies on the timestamp-invariance property of the observation encoder. It is unclear whether this assumption holds broadly across different flow-matching VLA implementations or other VLA architectures. Could the authors clarify the scope under which this assumption remains valid?

4. The experimental evaluation focuses on a specific family of flow-matching VLA models ($\Pi_0 / \Pi_{0.5}$). How well does the proposed streaming architecture generalize to other VLA architectures, such as diffusion-based or autoregressive policies? In addition, although the paper is motivated by real-time robotic control, the evaluation is primarily conducted in simulation benchmarks. Have the authors conducted any experiments on real robotic systems to validate the practical deployment benefits?

5. The proposed system integrates several known system-level techniques, including KV-cache reuse, asynchronous execution, and operator fusion. It would be helpful for the authors to clarify which component contributes most to the observed improvements and whether the key gains arise from the architectural redesign or from low-level engineering optimizations.

**Limitations:**

yes

**Strengths And Weaknesses:**

**Strengths:**

1. The paper addresses the practical challenge of deploying VLA models for real-time robotic control, where inference latency can significantly degrade control performance.

2. The proposed architecture integrates several system-level optimizations, including partitioned attention caching, asynchronous pipelines, and memory-efficient buffering, forming a coherent streaming inference framework.

3. Experiments evaluate multiple aspects of system performance (latency, reaction latency, stall rate, memory usage), providing useful insights into the practical benefits of the proposed system.

**Weaknesses:**

1. The paper repeatedly claims mathematical equivalence to full-batch inference, but the experimental validation appears limited to the partitioned attention mechanism. It remains unclear whether this guarantee holds for the entire streaming pipeline under asynchronous execution and future-state conditioning.

2. The asynchronous policy stream relies on a future-state predictor to compensate for latency, but the paper provides little empirical analysis of its robustness or the impact of prediction errors on policy performance.

3. The proposed approach relies on the timestamp-invariance property of the observation encoder and is evaluated primarily on a specific family of flow-matching VLA models. It remains unclear how well the method generalizes to other VLA architectures such as diffusion-based or autoregressive policies.

---

> ### Author Rebuttal · Authors · 2026-03-31
>
> We thank the reviewer for the thoughtful questions. We address each with new experimental evidence and clarifications.
>
> ## Q1: Equivalence Scope
>
> We apologize that our previous wording over-scoped the guarantee. In the revision, we will make explicit that the formal equivalence claim applies to **Partitioned Attention for fixed inputs and a fixed observation window**, not to the entire asynchronous pipeline. Specifically, we will update: Abstract, Section 2.3 G1, Section 3.1, Appendix A.1, and Conclusion to replace "Reflex is mathematically equivalent to full-batch inference" with "Partitioned Attention is mathematically equivalent to full-batch attention for a fixed observation window and fixed inputs."
>
> Asynchronous scheduling and future-state prediction intentionally change the input state and are therefore evaluated empirically rather than covered by the formal exactness claim. Mixed precision does not enlarge the equivalence claim either; it is a numerical-stability intervention. Our exactness statement is always relative to the same inputs under the same computational setting.
>
> The BF16 stress test (E6) addresses long-horizon stability separately:
>
> | Variant | Max Stable Steps | NaN/Inf Events | Added Latency (ms) |
> |---------|-----------------:|---------------:|--------------------:|
> | BF16 baseline | 120-220 | frequent after collapse | 0.0 |
> | BF16 + FP32 norm-only | 700-1200 | rare | +0.8 |
> | BF16 + AdaRMSNorm | **>2000** | **none observed** | **+0.4** |
>
> AdaRMSNorm maintains numerical stability for >2000 steps vs baseline collapse at 120-220 steps.
>
> ## Q2: Predictor Sensitivity in Dynamic Scenarios
>
> On Kinetix with controlled delay:
>
> | Delay Steps | Reflex (%) | w/o Predictor (%) | Async-Naive (%) |
> |:-----------:|-----------:|-------------------:|----------------:|
> | 1 | 78 | 76 | 73 |
> | 2 | 74 | 68 | 61 |
> | 4 | 69 | 58 | 47 |
>
> At delay=4, the predictor adds **+11pp** over Reflex w/o Predictor and **+22pp** over Async-Naive.
>
> The real-robot results on **AgileX PiPer** (3 tasks, 180 episodes; full table in R1-Q2) further validate predictor robustness under physical perturbation: Reflex improves success by +11–17pp over Sync with 0% stall and 101–110ms reaction latency. Notably, Async-Naive underperforms even Sync on Dynamic Recovery (32% vs 38%) — a clear indication that the predictor is not merely a latency optimization but a correctness-relevant component when observations change rapidly.
>
> ## Q3/Q4: Generalization to Other VLA Architectures
>
> On **X-VLA-0.9B**:
>
> | Benchmark | Method | Inf. Latency (ms) | Rxn. Latency (ms) | Memory (GB) | Success (%) |
> |-----------|--------|------------------:|------------------:|------------:|------------:|
> | LIBERO Avg | Sync | 82.4 | 104.6 | 5.8 | 92.8 |
> | LIBERO Avg | Async-Naive | 49.8 | 67.2 | 5.0 | 92.0 |
> | LIBERO Avg | Reflex | **34.7** | **43.5** | **4.5** | **93.5** |
> | Simpler-WidowX | Sync | 86.1 | 108.8 | 5.9 | 54.2 |
> | Simpler-WidowX | Async-Naive | 52.6 | 70.1 | 5.1 | 51.6 |
> | Simpler-WidowX | Reflex | **36.8** | **45.2** | **4.6** | **56.0** |
>
> On **WidowX** real robot:
>
> | Platform | Method | Success (%) | Reaction Latency (ms) |
> |----------|--------|------------:|----------------------:|
> | WidowX | Sync | 81 | 112 |
> | WidowX | Reflex | **85** | **48** |
>
> These results correspond to **2.4x** inference speedup, **22%** memory reduction, and success parity or improvement.
>
> Scope: Reflex applies when the observation encoder does not take $t_k$ as input. We validate this on Pi0, Pi0.5, SmolVLA, and X-VLA. Unified DiT-style architectures where timestep conditioning enters the vision encoder (e.g., AdaLN in RDT-1B) violate this condition and are out of scope. Autoregressive controllers are instead a simpler case for the cache-partitioning component, because they do not suffer from timestep-induced cache invalidation; our empirical focus here is the harder flow-matching setting.
>
> ## Q5: Component Contributions
>
> The incremental ablation on Pi0.5 (LIBERO Avg) is:
>
> | Method | Inf. Latency (ms) | Rxn. Latency (ms) | Success (%) |
> |--------|------------------:|------------------:|------------:|
> | Sync | 135.2 | 156.5 | 78.0 |
> | + Partitioned Attn | 61.5 | 151.8 | 78.1 |
> | + AdaRMSNorm | 61.9 | 151.9 | 78.2 |
> | + Operator Fusion | 52.4 | 149.6 | 78.3 |
> | + Async Pipeline | 52.6 | 92.4 | 79.2 |
> | Full Reflex | 52.4 | 82.5 | 79.5 |
>
> The long-horizon BF16 stability results are:
>
> | Method | Max Stable Steps | NaN/Inf Events |
> |--------|----------------:|---------------:|
> | Sync (BF16 baseline) | 120-220 | frequent |
> | + AdaRMSNorm | >2000 | none observed |
>
> Partitioned Attention provides the largest inference speedup (135.2→61.5ms, 2.2x), AdaRMSNorm provides stability, and the async pipeline halves reaction latency (151.9→82.5ms). Each component is independently justified.

---

> > ### Author Rebuttal · Reviewer_cjJe · 2026-04-02
> >
> > The rebuttal provides clear responses and additional experimental evidence addressing my concerns. I appreciate the clarification of the equivalence claim and the added evaluations on predictor robustness and real-world deployment.

---

> > > ### Author Response · Authors · 2026-04-03
> > >
> > > Thank you very much for your positive follow-up and for recognizing that our rebuttal has adequately addressed your concerns. We truly appreciate your time and thoughtful evaluation. Given that you indicated the concerns are now fully resolved, we would be sincerely grateful if you could consider updating your score to better reflect your current assessment of our paper.

---

### Official Review · Reviewer_w3Kc · 2026-03-12

**Soundness:** 3
**Presentation:** 3
**Significance:** 4
**Originality:** 3
**Overall Recommendation:** 4
**Confidence:** 3

**Summary:**

This paper considers a challenge in deploying Vision-Language-Action (VLA) models for real-time robotic control. The authors outline the key problem that current VLA policies generate actions through iterative denoising, which introduces latency and prevents standard transformer optimizations such as KV-caching because timestep conditioning invalidates cached representations. To address this, the paper proposes Reflex, a streaming inference architecture designed to enable real-time execution of flow-matching VLA models. The method exploits a property that perception encoders are independent of denoising timesteps, allowing the model context to be partitioned into static, sliding, and dynamic regions. This partitioned attention mechanism enables O(1) cache updates while maintaining outputs identical to full-batch inference. The system also introduces AdaRMSNorm, an adaptive normalization layer to prevent numerical instability during long streaming runs, and an asynchronous pipeline that overlaps visual encoding and action generation. Experiments on LIBERO and Kinetix demonstrate a 2.58× inference speedup, stable 50 Hz control loops, and up to 54% reduction in reaction latency, while maintaining or improving task success rates compared to baseline implementations.

**Compliance With Llm Reviewing Policy:**

Affirmed.

**Key Questions For Authors:**

1. The proposed framework is demonstrated on pi0 and pi0.5 models. How the proposed method could be applied to different VLA models, e.g., GR00T N1 or GR00T N1.5? Since GR00T use larger VLM backbone, the latency problem would be much more crucial here.  In particular, would the partitioned attention mechanism could be easily generalized to such models?

2. Reflex introduces several components simultaneously (partitioned attention, AdaRMSNorm, asynchronous pipeline, operator fusion). Could the authors provide additional ablations to quantify the individual contribution of each component to latency reduction and task success?

3. The experiments are conducted on benchmark environments such as LIBERO and Kinetix. Have the authors evaluated Reflex on real-world robot problems, or could the author analyze the anticipated challenges that might affect the inference pipeline? I believe since the paper is about optimizing the inference on system-level, it is crucial to show off the real-world deployment.

**Limitations:**

yes

**Strengths And Weaknesses:**

Strengths

One major strength of the paper is its system-level perspective on the control-inference gap in VLA models. Rather than focusing solely on accelerating model components, the authors reformulate the problem by architectural scheduling and streaming inference. The partitioned attention mechanism is sound, as it leverages the separation between timestep-invariant perception encoders and timestep-dependent denoising components. This insight allows Reflex to reuse cached representations while preserving equivalence with full recomputation. Also, the paper demonstrates the strength in comprehensive system design that integrates algorithmic, architectural, and hardware-level optimizations. The paper proposes several complementary components, e.g., AdaRMSNorm for numerical stability, asynchronous pipeline scheduling, operator fusion, and ring-buffer memory management, that enable stable 50 Hz control loops. The experimental evaluation is also relatively thorough, covering both static manipulation tasks (LIBERO) and dynamic environments (Kinetix). The reported results demonstrate consistent improvements in latency, memory efficiency, and task success rates while maintaining identical predictions to the full-batch baseline.

Weaknesses

First, many components of the Reflex system are introduced simultaneously, making it difficult to isolate the contribution of each element beyond the limited ablations presented. Second, while the system achieves significant latency reductions, the evaluation is largely conducted on simulated or benchmark environments, leaving open questions about robustness in real-world robotic deployments with sensor noise, delays, and hardware constraints. Additionally, the theoretical justification of the method could be presented more rigorously. While the architectural observation that perception encoders are independent of denoising timesteps is intuitive, the manuscript provides limited formal analysis on how it generalizes across different VLA architectures. Finally, the method is tailored to flow-matching policies, which may limit its applicability to alternative action-generation paradigms such as autoregressive or diffusion-based controllers.

---

> ### Author Rebuttal · Authors · 2026-03-31
>
> We thank the reviewer for the constructive feedback. We address each question with new evidence.
>
> ## Q1: Generalization to Other VLA Architectures (e.g., GR00T N1)
>
> In the revision we will formalize a simple applicability criterion:
>
> **Generalized Timestep-Invariance Condition (GTIC).** Reflex applies whenever a VLA can be decomposed into a perception module $\mathrm{Enc}(\cdot)$ and an action module $\mathrm{Act}(\cdot, t)$ with $\partial \mathrm{Enc}/\partial t = 0$, i.e., the observation encoder does not receive the denoising timestep.
>
> This condition is architectural rather than specific to Pi0. **GR00T N1.5** satisfies it: Eagle 2.5 produces observation embeddings, while timestep conditioning enters the DiT-based flow-matching action head. We implemented Reflex on GR00T N1.5 with only configuration changes.
>
> The new GR00T N1.5 results directly answer the reviewer's question. On LIBERO-Spatial, Partitioned Attention remains exact: full recomputation gives MSE=0.00 / success=84.8, naive caching gives MSE>1.0 / success<15.0, and Reflex gives MSE=0.00 / success=85.2. On RTX 4090 / LIBERO, latency drops from 188.4ms to 66.0ms (2.85x) with 25% memory saving. The larger Eagle 2.5 backbone makes standard vision prefill more expensive (72.4ms vs. 42.1ms on Pi0.5), and Reflex reduces it to 13.8ms. This confirms your intuition that larger VLM backbones make the latency problem more acute, and therefore amplify Reflex's benefit rather than reducing it.
>
> | Model      | Std. Latency | Reflex Latency | Speedup | Memory Saving |
> | ---------- | -----------: | -------------: | ------: | ------------: |
> | Pi0.5      |      135.2ms |         52.4ms |   2.58x |           27% |
> | Pi0        |      183.6ms |         67.2ms |   2.73x |           24% |
> | GR00T N1.5 |      188.4ms |         66.0ms |   2.85x |           25% |
>
> For autoregressive controllers such as OpenVLA, the cache-partitioning component reduces to the easier pinned-prefix/sliding-window case because there is no timestep-induced cache invalidation. We also observe the same trend on SmolVLA and X-VLA; in the revision we will surface those cross-family results more prominently.
>
> | Setting              | Method | Inf. |  Rxn. | Success |
> | -------------------- | ------ | ---: | ----: | ------: |
> | X-VLA LIBERO         | Sync   | 82.4 | 104.6 |    92.8 |
> | X-VLA LIBERO         | Reflex | 34.7 |  43.5 |    93.5 |
> | X-VLA Simpler-WidowX | Sync   | 86.1 | 108.8 |    54.2 |
> | X-VLA Simpler-WidowX | Reflex | 36.8 |  45.2 |    56.0 |
> | WidowX robot         | Sync   |    - |   112 |      81 |
> | WidowX robot         | Reflex |    - |    48 |      85 |
>
> ## Q2: Component Ablation
>
> New incremental ablation on Pi0.5 (LIBERO Avg) isolates the role of each component. Partitioned Attention provides the largest inference-latency reduction: **135.2→61.5ms**. AdaRMSNorm does not target latency; it targets stability, extending BF16 streaming from **120-220** to **>2000** stable steps with no observed NaN/Inf. Operator fusion gives the final single-stream speedup: **61.9→52.4ms**. Async execution and future-conditional overlap scheduling target reaction latency: **151.9→82.5ms**. The intermediate async-only stage reaches **92.4ms**; the final **92.4→82.5ms** gain comes from overlap scheduling eliminating tail idle time. Success remains stable throughout (**78.0→79.5**), indicating that the gains are primarily systems-side rather than accuracy-through-tuning.
>
> | Stage        |  Inf. |  Rxn. | Success |
> | ------------ | ----: | ----: | ------: |
> | Sync         | 135.2 | 156.5 |    78.0 |
> | + PartAttn   |  61.5 | 151.8 |    78.1 |
> | + AdaRMSNorm |  61.9 | 151.9 |    78.2 |
> | + Fusion     |  52.4 | 149.6 |    78.3 |
> | + Async      |  52.6 |  92.4 |    79.2 |
> | Full Reflex  |  52.4 |  82.5 |    79.5 |
>
> ## Q3: Real-Robot Evaluation
>
> We added an **AgileX PiPer** evaluation with **3 tasks × 3 methods × 20 episodes = 180 episodes**, using the same frozen checkpoint and hyperparameters as in simulation. Reflex improves success by **+11pp** on Pick-Place, **+14pp** on Articulated, and **+17pp** on Dynamic Recovery over Sync, while reducing reaction latency to **101-110ms** and eliminating stalls entirely (**0%** vs. **100%** for Sync). Async-Naive still stalls on **20-33%** of episodes and underperforms Sync on Dynamic Recovery, showing that lower latency alone is insufficient without state alignment. These experiments are intended as deployment validation rather than a full sim-to-real study, but they directly confirm the system-level benefit on physical hardware.
>
> | Task             | Success Gain vs. Sync | Reaction Latency | Stall |
> | ---------------- | --------------------: | ---------------: | ----: |
> | Pick-Place       |                 +11pp |            101ms |    0% |
> | Articulated      |                 +14pp |            104ms |    0% |
> | Dynamic Recovery |                 +17pp |            110ms |    0% |

---

> > ### Author Rebuttal · Reviewer_w3Kc · 2026-04-06
> >
> > Thank you for the rebuttal. I remain positive on the paper, and maintain the score as Weak accept.

---

> > > ### Author Response · Authors · 2026-04-07
> > >
> > > Thank you very much for your response and for your careful consideration. We sincerely appreciate your time and feedback.

---

### Official Review · Reviewer_4vkR · 2026-03-13

**Soundness:** 2
**Presentation:** 2
**Significance:** 2
**Originality:** 2
**Overall Recommendation:** 3
**Confidence:** 4

**Summary:**

The paper introduces Reflex, a streaming inference framework aimed at reducing the latency of flow matching-based Vision-Language-Action (VLA) models. The authors identify that timestep conditioning during iterative denoising invalidates standard KV-caching, causing severe blocking latency in real-time control . To address this, Reflex proposes a "Partitioned Attention" mechanism to isolate timestep-invariant visual features for constant-time cache updates , an AdaRMSNorm layer to prevent BFloat16 numerical underflow , and asynchronous execution coupled with CUDA operator fusion . The system is evaluated on the LIBERO and Kinetix simulation benchmarks, reporting a 2.58x inference speedup and stable 50Hz control .

**Compliance With Llm Reviewing Policy:**

Affirmed.

**Ethical Review Concerns:**

Thanks for the additional real-robot experiments on AgileX PiPer, which address my earlier concern about the lack of hardware evaluation. I am raising my score from 2 to 3. That said, my concern about novelty remains. I still feel the paper falls slightly short in this regard, as the partitioned attention mechanism essentially adapts existing KV-cache segmentation techniques from LLM serving to the VLA setting.

**Final Justification:**

Thanks for the additional real-robot experiments on AgileX PiPer, which address my earlier concern about the lack of hardware evaluation. I am raising my score from 2 to 3. That said, my concern about novelty remains. I still feel the paper falls slightly short in this regard, as the partitioned attention mechanism essentially adapts existing KV-cache segmentation techniques from LLM serving to the VLA setting.

**Key Questions For Authors:**

1. The authors present "Partitioned Attention" as a key conceptual breakthrough to maintain cache validity . While the observation that the visual backbone is timestep-invariant ($\partial Enc/\partial t_k = 0$)  is a practical engineering insight, the proposed solution is mathematically trivial. Equation (1) simply demonstrates the standard concatenation of block-wise KV-caches (a static prefix, a sliding window, and a recomputed suffix) . This specific memory management mechanism—segmenting context into pinned prefixes and sliding histories—is already ubiquitous in modern LLM serving engines (e.g., Prefix Caching). Adapting this established systems-level trick to isolate timestep-dependent representations from timestep-invariant ones is a clever engineering implementation. However, re-labeling basic KV-cache concatenation as a novel "Partitioned Attention" overstates the methodological contribution and lacks the mathematical depth required to be considered a fundamental algorithmic advancement in generative modeling.

2. Despite the paper's title explicitly claiming "Real-Time Vision-Language-Action Control" and its strong motivational emphasis on solving physical deployment bottlenecks (such as 50Hz continuous control and reaction latency) , the quantitative evaluation is strictly confined to simulation benchmarks (LIBERO and Kinetix) . While there is a brief, anecdotal mention of a Franka Panda setup , the paper completely lacks any systematic real-world hardware experiments, physical task success rates, or rigorous Sim-to-Real transfer analysis. For a framework whose entire premise is predicated on bridging the gap between computation and physical execution in dynamic environments, relying exclusively on simulated environments severely diminishes the persuasiveness and credibility of the empirical claims.

3. To bridge the temporal gap introduced by asynchronous execution, the authors rely on a "Future-Conditional State Prediction" where they simply substitute the stale sensor reading with the previously commanded action ($s_{t+\Delta} \approx a_t^{cmd}$) . This first-order approximation is an extremely simplistic heuristic. It completely bypasses the complex, non-linear transition dynamics of the real physical world. A principled machine learning approach would involve learning a predictive world model or formulating a temporally aligned flow matching ODE. Relying on such a rudimentary heuristic further underscores the paper's lack of algorithmic depth.

**Strengths And Weaknesses:**

1. The paper addresses a highly practical deployment bottleneck. The memory management design, such as using a pre-allocated Ring Buffer to eliminate memory fragmentation , demonstrates solid system-level engineering.

2. The empirical profiling of inference bottlenecks and the demonstration of high-frequency control on the dynamic Kinetix benchmark are well-executed.

3. Please refer to the Questions section below for a detailed discussion of the paper's weaknesses.

---

> ### Author Rebuttal · Authors · 2026-03-31
>
> We thank the reviewer for the detailed critique. We have completed 6 supplemental experiments that directly address each concern.
>
> ## Q1: Novelty of Partitioned Attention
>
> We would like to point out that our core contribution is a **systems/runtime contribution**, not a new generative-modeling algorithm. Our claim is not that block-wise KV concatenation itself is novel, but that we identify a sufficient condition for exact cache reuse in timestep-conditioned VLA inference, show empirically that naive reuse degrades both output fidelity and policy success, and build a streaming VLA serving design around that condition. In the revision we will reframe Partitioned Attention as a structure-aware cache partitioning scheme for streaming VLA serving, with a clearer comparison to prefix/paged caching. **ICML has repeatedly welcomed system/efficiency papers on training and inference optimization [R1][R2][R3]; we will position Reflex in that tradition rather than as a new generative-modeling algorithm.**
>
> LLM prefix caching reuses a fixed prefix across *requests*. Here, the denoising loop runs *within* one request, and the action expert receives a fresh timestep embedding at each step. Action queries attend to observation K/V, but the observation encoder never attends to action tokens or receives timestep signals. Hence $\partial \text{Enc}/\partial t_k = 0$, so observation K/V remain unchanged across denoising steps and partitioned attention is exactly equivalent to full-batch attention for fixed inputs.
>
> Without this condition, naive cache reuse feeds timestep-mismatched representations into the action expert. Table 3 confirms this: Naive Cache yields MSE=1.42 and 12.5% success, while our partition achieves MSE=0.00 and 85.4% success — numerically identical to full-batch attention within measured precision for the same fixed inputs.
>
> Refs:
>
> [R1] Moccasin: Efficient Tensor Rematerialization for Neural Networks, ICML 2023.
>
> [R2] InferCept: Efficient Intercept Support for Augmented Large Language Model Inference, ICML 2024.
>
> [R3] HexGen: Generative Inference of Large Language Model over Heterogeneous Environment, ICML 2024.
>
> ## Q2: Real-Robot Evaluation
>
> We added an **AgileX PiPer** evaluation: 3 tasks × 3 methods × 20 episodes = 180 total episodes, using the same checkpoint and hyperparameters as in simulation. ± values are standard deviations across episodes.
>
> | Task | Method | Success (%) | Reaction Latency (ms) | Stall (%) |
> |------|--------|------------:|----------------------:|----------:|
> | Pick-Place | Sync | 65±8 | 191 | 100 |
> | Pick-Place | Async-Naive | 62±8 | 148 | 20±8 |
> | Pick-Place | Reflex | **76±7** | **101** | **0** |
> | Articulated | Sync | 52±9 | 198 | 100 |
> | Articulated | Async-Naive | 48±11 | 155 | 32±9 |
> | Articulated | Reflex | **66±9** | **104** | **0** |
> | Dynamic Recovery | Sync | 38±8 | 205 | 100 |
> | Dynamic Recovery | Async-Naive | 32±8 | 160 | 33±9 |
> | Dynamic Recovery | Reflex | **55±9** | **110** | **0** |
>
> Reflex improves real-robot success by **+11pp** (Pick-Place), **+14pp** (Articulated), and **+17pp** (Dynamic Recovery) over Sync, with **0% stall** (vs 100% Sync, 20–33% Async-Naive) and 101–110ms reaction latency across all tasks.
>
> Notably, Async-Naive underperforms Sync on Dynamic Recovery (32% vs 38%) — stale observations cause the robot to commit to an outdated trajectory, which is worse than waiting. Reflex's predictor enables re-planning on predicted current state, eliminating this regression and yielding the largest gain on the hardest task.
>
> We do not intend to present this as a full sim-to-real transfer study; rather, these results provide the systematic physical validation that was missing from the submission.
>
> ## Q3: Future-State Predictor
>
> We agree that this is a lightweight short-horizon heuristic rather than a learned world model. However, the predictor novelty is **not** the algorithmic claim of Reflex. Our emphasis is the **end-to-end streaming system**: Partitioned Attention, asynchronous execution, operator fusion, ring-buffered memory management, and a lightweight state-alignment module working together to remove blocking latency in VLA serving. In that context, the predictor is intentionally simple because it must run inside a tight real-time loop, and the real-robot Dynamic Recovery result in Q2 shows that this choice is already sufficient to avoid the stale-state failure mode that causes Async-Naive to underperform even Sync. We will therefore present it as a systems component, not as a new predictive-modeling algorithm.

---

> > ### Author Rebuttal · Reviewer_4vkR · 2026-04-03
> >
> > Thanks for the author's reply. I will make my decision later based on the discussion.

---

> > > ### Author Response · Authors · 2026-04-07
> > >
> > > We thank the reviewer for the additional comment and we fully understand that the final judgment will be made based on the discussion. As the discussion period is now approaching its end, we sincerely hope the reviewer may take into account our rebuttal together with the other discussion records in making the final evaluation.

---

### Decision · Program_Chairs · 2026-04-30

**Decision:**

Accept (regular)

**Comment:**

This paper introduces a streaming inference framework for flow-matching VLA control. It combines exact cache reuse for timestep-invariant context, stable mixed-precision streaming, and asynchronous execution to reduce reaction latency. Initial reviews raised concerns about the novelty of the partitioned-attention design, the lack of stronger real robot experiments, and whether the mathematical-equivalence and async claims were fully supported. In the rebuttal, the authors clarified that the main contribution is an efficiency system/runtime solution rather than a new generative-modeling algorithm. They also added real-robot results and additional experiments on the equivalence claim and predictor robustness, and resolved most of the technical concerns. After discussion, `27eJ` raised the score to 4, `w3Kc` remained Weak Accept, and both `cjJe` and `4vkR` acknowledged the added evidence while maintaining Weak Reject, probably because they still viewed the contribution as primarily system-oriented.

While AC does not view novelty as a concerning issue here, the full async pipeline relies on a lightweight future-prediction module based on a first-order approximation, which is better viewed as a latency-compensation heuristic than as a strict end-to-end equivalence result of the whole system. Thus, the exact cache-reuse component is more convincing than the full async story. Given the otherwise positive discussion profile and the fact that most concerns were addressed in rebuttal, AC recommends Weak Accept.